# Beyond PARP Inhibitors in Advanced Breast Cancer Patients with Germline *BRCA1/2* Mutations: Focus on CDK4/6-Inhibitors and Data Review on Other Biological Therapies

**DOI:** 10.3390/cancers15133305

**Published:** 2023-06-23

**Authors:** Marta Nerone, Lorenzo Rossi, Rosaria Condorelli, Vilma Ratti, Fabio Conforti, Antonella Palazzo, Rossella Graffeo

**Affiliations:** 1Service of Medical Oncology, Oncology Institute of Southern Switzerland, Ente Ospedaliero Cantonale (EOC), 6500 Bellinzona, Switzerland; lorenzo.rossi@eoc.ch (L.R.); rosaria.condorelli@eoc.ch (R.C.); vilma.ratti@eoc.ch (V.R.); rossella.graffeogalbiati@eoc.ch (R.G.); 2Oncology Unit, Humanitas Gavazzeni, 24125 Bergamo, Italy; fabio.conforti@gavezzeni.it; 3Comprehensive Cancer Center, Fondazione Policlinico Universitario Agostino Gemelli, IRCCS, 00168 Rome, Italy; antonella.palazzo@policlinicogemelli.it

**Keywords:** advanced breast cancer, germline *BRCA1/2* mutations, pathogenic/likely pathogenic variants, CDK4/6 inhibitors, Pi3Ka inhibitors, HER2-low

## Abstract

**Simple Summary:**

The treatment landscape for advanced breast cancer (BC) has been expanding due to the development of biological drugs with demonstrated efficacy across all breast cancer subtypes. The role of poly adenosine diphosphate–ribose polymerase (PARP) inhibitors in advanced BC is well established for the treatment of germline *BRCA1/2* mutated (g*BRCA1/2*m) HER2-negative BC, regardless of hormone receptor (HR) expression. Nevertheless, there is very little data on the efficacy of other drugs in the event of disease progression in patients with g*BRCA1/2*m BC. This review provides an update on the efficacy of biological drugs, approved in the US and Europe, for the treatment of advanced BC in this patient population.

**Abstract:**

We explored the outcomes of germline *BRCA1/2* pathogenic/likely pathogenic variants (PVs/LPVs) in the endocrine-sensitive disease treated with first-line standard of care cyclin-dependent kinase 4/6 (CDK4/6) inhibitors. Three studies retrospectively showed a reduction in the overall survival (OS) and progression-free survival (PFS) in g*BRCA1/2*m patients compared to both the germinal *BRCA1/2* wild type (g*BRCA1/2*wt) and the untested population. Regarding the efficacy of PI3Kα inhibitors, there are no subgroups or biomarker analyses in which germinal BRCA status was explored. However, the biological interactions between the PIK3CA/AKT/mTOR pathway and *BRCA1/2* at a molecular level could help us to understand the activity of these drugs when used to treat BC in *BRCA1/2* PVs/LPVs carriers. The efficacy of trastuzumab deruxtecan (T-DXd), an antibody–drug conjugate (ADC) targeting HER2 for HER2-low and HER2-positive (HER2+) BC, has been increasingly described. Unfortunately, data on T-DXd in HER2+ or HER2-low metastatic BC harboring germinal *BRCA1/2* PVs/LPVs is lacking. Including germinal *BRCA1/2* status in the subgroup analysis of the registration trials of this ADC would be of great interest, especially in the phase III trial DESTINY-breast04. This trial enrolled patients with HER2-negative (HER2−) and both HR+ and HR− metastatic disease, which can now be categorized as HER2-low. The HER2-low subgroup includes tumors that were previously classified as triple negative, so it is highly likely that some women were germline *BRCA1/2* PVs/LPVs carriers and this data was not reported. Germline *BRCA1/2* status will be available for a higher number of individuals with BC in the near future, and data on the prognostic and predictive role of these PVs/LPVs is needed in order to choose the best treatment options.

## 1. Introduction

BC is the most common malignancy in females in Western countries, and advanced breast cancer (ABC) still represents the leading cause of cancer-related death in women [1,2]. Over the past decades, the treatment algorithm for the advanced disease was based essentially on endocrine therapy (ET) for the hormone receptor-positive (HR+) ABC, systemic chemotherapy for triple-negative breast cancer (TNBC), and chemotherapy plus anti-human epidermal growth factor receptor 2 (HER2) target therapy for the HER2+ disease.

Advances in the treatment landscape for ABC are significant nowadays thanks to the development and approval of several biological drugs that have proven efficacy across all different subtypes of BC [3]. Therefore, today we face a wide range of therapeutic possibilities for patients, and the choice of one treatment strategy over another might be debatable.

Germline pathogenic/likely pathogenic (P/LP) variants in *BRCA1/2* genes account for about 2.8–7% of all BCs [4,5], and their prevalence in metastatic BC is estimated to be around 6% [6]. Those genes are important tumor suppressors implicated in the homologous recombination DNA repair mechanisms. Women who inherit a PV/LPV in one of the *BRCA* genes have a 45–87% lifetime risk of developing BC [7,8].

The role of poly adenosine diphosphate–ribose polymerase (PARP) inhibitors in ABC with germline *BRCA* PV/LPVs is well established, and two PARP inhibitors (PARPi), olaparib and talazoparib, are currently approved for the treatment of metastatic g*BRCA1/2*m HER2− BC, irrespective of HR expression [9,10]. The data on the PFS and overall response rate (ORR) are encouraging; however, subsequent data on the OS did not show a statistically significant improvement over chemotherapy [11,12]. The need for additional lines of therapy in the event of disease progression or in the case of PARPi unavailability persists, and data on the efficacy of other drugs in the population with germline PVs/LPVs in *BRCA1/2* genes are required. Information derived from the germinal status could be helpful when selecting first-line chemotherapy in order to offer the treatment associated with the best response rates in BC *BRCA1/2* PV carriers in an earlier setting.

As far as cytotoxic chemotherapy is concerned, a superior response rate with carboplatin has been demonstrated in tumors characterized by a loss of function in *BRCA1/2* genes [13]; however, data on the efficacy of new biological agents according to the *BRCA1/2* status are still missing.

The objective of this review is to collect data regarding the preclinical and clinical activity of new-generation drugs across different BC intrinsic subtypes caused by germline PVs in *BRCA1/2* genes, with the aim of helping to optimize patient and treatment selection while trying to understand the molecular pathways that lead to greater or lesser therapy response.

## 2. Hormone Receptor-Positive Breast Cancer

### 2.1. CDK4/6 Inhibitors

In this review, we explore the outcomes of germline *BRCA1/2* P/LP variants in endocrine-sensitive BC treated with oral agents that inhibit CDK 4/6. Recently, therapeutic strategies for HR+ and HER2− metastatic BC have been moving in favor of endocrine therapies combined with a CDK 4/6 inhibitor (CDK4/6i).

In several randomized phase III trials (PALOMA-2, MONALEESA-2, MONALEESA-7 and MONARCH-3), the combination of a nonsteroidal aromatase inhibitor (NSAI) with CDK4/6is (palbociclib, ribociclib, or abemaciclib) as a first-line strategy in the treatment of HR+ and HER2-negative metastatic BC showed a significantly longer PFS [14,15,16,17]. More recently, a significant OS benefit was reached with ribociclib as a first-line combined treatment for post- and premenopausal patients, with a relative reduced risk of death of 29% (OS HR: 0.71; 95% CI: 0.54 to 0.95; *p*-value = 0.00973) and 24% (OS HR 0.76; 95% CI: 0.63–0.93; two-sided *p* = 0.008), respectively [16,18]. A significant clinical benefit has also been demonstrated in the second-line setting with the association of fulvestrant and a CDK4/6i in terms of both PFS and reduced risk of death [19,20,21].

In these phase III trials investigating the addition of CDK4/6i to endocrine therapy in patients with HR+ HER2− metastatic BC, a substantial benefit was found; however, the g*BRCA1/2* status was not reported. 

In a retrospective analysis, 2968 patients with HR+/HER2− metastatic BC received a CDK4/6i, and 859 (28.9%) had known gBRCA status, of whom 9.9% were g*BRCA1/2*m. A shorter time to first subsequent therapy or death (TFST) (stratified HR: 1.24; 95% CI: 0.96–1.59) and OS (stratified HR 1.50; 95% CI: 1.06–2.14) was observed in patients carrying germline *BRCA1/2* P/LP variants compared to the g*BRCA1/2*wt group [22].

Another retrospective cohort study evaluated 217 patients with HR+ HER2− ABC receiving a CDK4/6i in combination with ET, 6.9% of whom harbored g*BRCA1/2* PVs/LPVs. In these patients, a significantly shorter median PFS (10.2 months, 95% CI: 5.7 to 14.7) was observed compared to those with g*BRCA1/2*wt tumors (15.6 months, 95% CI: 7.8 to 23.4) and the untested individuals (17.9 months, 95% CI: 12.9 to 22.2; *p* = 0.002) [23]. 

Consistently, a retrospective study conducted in Korea showed that the presence of a germline genetic alteration in *BRCA1/2* is associated with an inferior PFS in ABC patients treated with palbociclib plus ET, both compared to g*BRCA1/2*wt patients (9.0 months vs. not reached, *p* = 0.031) and the untested population (9.0 months vs. 33.0 months, *p* = 0.001) [24].

These results should be considered with caution due to the small proportion of patients who underwent genetic testing and the small sample of women harboring germline P/LP variants.

In contrast, a pooled analysis of biomarkers predictive of response or resistance to ribociclib was recently presented as a subgroup analysis of the MONAALESA trials. This analysis was performed on circulating tumor DNA (ctDNA) by the next-generation sequencing (NGS) method for the presence of genetic alterations in patients with metastatic BC treated with ribociclib in combination with endocrine therapy versus ET alone. The presence of *BRCA1/2* P/LP variants correlated with an improved PFS when ribociclib was administered. Different results were found with the somatic *ATM* variant, suggesting a potential predictive biomarker of resistance to ribociclib [25]. However, this finding refers to somatic ctDNA results, and it is not known whether they have been compared with the patients’ germline status. Therefore, the information that germline mutations in these genes lead to a similar disease course can only be extrapolated with caution.

In agreement with this finding, two clinical cases of g*BRCA1/2*m patients and refractory HR+ metastatic BC reported a durable response to the combination of palbociclib and endocrine therapy [26].

Evidence suggests that CDK4/6 inhibitors in combination with endocrine therapy can induce a synthetic lethal effect on *BRCA1/2* mutant and HR+ tumor cells. *BRCA*1/2 genes are involved in the regulation of DNA repair mechanisms but are also strategic for HR expression and function. It has been extensively described that wild-type *BRCA1* exerts inhibition on ER alpha (ERα). In the presence of *BRCA1* P/LP variants, this inhibition is abolished or reduced, providing the rationale for higher ERα activity and, consequently, higher sensitivity to endocrine therapy. In addition, cell cycle arrest fails in the presence of germline *BRCA1/2* mutations, but CDK4/6 inhibitors could restore G1 arrest. The induction of G1 cell cycle arrest can be used to manipulate the activity of DNA repair pathways, particularly in homologous recombination-deficient (HRD) cells. In these cells, G1 cell cycle arrest can lead to increased activity of non-homologous end joining (NHEJ), resulting in genomic instability and apoptosis. In addition, it has been demonstrated that *BRCA1* binds to hypophosphorylated RB, which acts as an inhibitor of cell proliferation. When a *BRCA1* P/LP variant is present, this anti-proliferative control mechanism is lost. The activity of a CDK4/6i may, therefore, be essential to restore G1 arrest, preventing the cell from entering mitosis [27].

Other authors have also shown that *BRCA1* promotes cell cycle arrest and tumor growth suppression through the induction of the cyclin-dependent kinase 2 (CDK2) inhibitor p21 [28]. CDK2 is an essential kinase for CDK4/6 inhibition and high expression of cyclin E1 (CCNE1), which activates CDK2, which was associated with palbociclib resistance in the PALOMA-3 study [29].

Thus, while there is preclinical evidence that *BRCA1* loss of function may correlate with good clinical outcomes when CDK4/6i in combination with ET is used, retrospective studies have not shown these positive outcomes in clinical practice.

It is clear that the interaction between cyclin-dependent kinases, estrogen receptors and *BRCA* is extremely complex, and the aforementioned results require further validation. 

Clinical trials investigating personalized combination treatment for ABC patients with germline or somatic *BRCA1/2* PVs/LPVs are ongoing (NCT03685331).

The currently available therapeutic strategies for HR+ and HER2− metastatic BC are based on the use of endocrine combination therapies with CDK4/6i in the first and second line, regardless of the *BRCA1/2* gene status (Table 1 and Figure 1).

### 2.2. PIK3CA Inhibitors

The phosphoinositide 3-kinase (PI3k) pathway is a central oncogenic pathway that regulates cellular proliferation, metabolism, growth, survival and apoptosis. Thus, it has been widely investigated as a target in solid tumors [30,31]. In advanced breast cancer, the alpha-specific PI3-Kinase (PI3Kα) inhibitor alpelisib, in combination with endocrine therapy (fulvestrant), showed promising results in HR+ HER2− breast cancer with PIK3CA mutations, with a doubling in the median PFS (11.0 vs 5.7 months), as reported in the phase III SOLAR-1 trial [32]. However, no subgroup analysis of this trial was conducted in order to examine the outcome of patients harboring germline *BRCA1/2* mutations. However, the correlation between the PI3CA/AKT/mTOR pathway and *BRCA1/2* has been extensively explored in preclinical and clinical settings. A preclinical study demonstrated that PI3K inhibition in TNBC cells led to DNA damage, the downregulation of BRCA1/2 and an increase in PARP activity, indicating that cells undergoing PI3K suppression become more dependent on this DNA repair mechanism and are, therefore, susceptible to PARP inhibition [33]. The synergy between PI3K inhibitors and PARPi has been observed both in vitro and in vivo and in homologous recombination repair proficient (HRP) as well as in homologous recombination repair deficient (HRD) models [34]. Another proposed mechanism of synergism between PI3K inhibitors and PARPi is the reduction of the production of nucleotides required for DNA synthesis, secondary to PI3K inhibition with a consequent reduction of glyceraldehyde 3-phosphate (Ga3P), that is essential for the production of the ribose 5-phosphate required for the synthesis of DNA and RNA. This can result in DNA damage and increased dependence on DNA repair mechanisms, which can in turn potentially increase the vulnerability to PARP inhibition [35]. This is important because studies have also shown that increasing glycolysis reduces sensitivity to olaparib. Similarly, the blocking of glycolysis resensitizes tumors to PARPi [36].

With increasing knowledge regarding the mechanisms of interaction between PI3K inhibitors and PARPi, a phase 1b trial with alpelisib plus olaparib for patients with ABC was conducted. Interestingly, this study included patients with advanced TNBC as well as patients with g*BRCA1/2*m BC, regardless of the tumor subtype. This was a dose-escalation and a dose-expansion trial, and the recommended phase 2 dose (RP2D) had been previously reported in patients with high-grade serous ovarian cancer. The secondary endpoints were safety and ORR. Although the conclusions are limited due to the small number of patients, the combination demonstrated activity in a heavily pretreated population, with ORR and clinical benefit rates of 18% and 35%, respectively. The objective of this study was to demonstrate the efficacy of the combination for *BRCA1/2* wild-type TNBCs, but the benefit was also observed in germinal *BRCA1/2* mutants, although they were underrepresented [37].

Another aspect that needs to be underlined, partially extrapolated from the data of this oral combination in patients with platinum-resistant recurrent ovarian cancer, is the role of PI3K inhibitors in re-establishing the homologous recombination deficiency. HRR restoration is, in fact, one of the mechanisms of ovarian cancer resistance to PARPi. Adding alpelisib to PARPi in case of disease progression can help rebuild HRD, making cancer cells sensitive to PARPi again.

Larger prospective randomized trials and biomarker development are needed to identify patients who are most likely to benefit from this all-oral combination. If it was confirmed that in breast cancer the addition of alpelisib can determine a re-sensitization to the PARPi, the olaparib plus alpelisib combination could be part of the treatment algorithm for g*BRCA1/2*m ABC, especially for those patients previously treated with olaparib who then progressed.

## 3. Triple-Negative Breast Cancer

### 3.1. Immune Checkpoint Inhibitors

Both the *BRCA1* and *BRCA2* genes encode proteins that play an essential role in maintaining genome integrity, primarily through their contribution to homologous recombination and in double-strand DNA break repair. *BRCA1/2* mutation-associated BCs have been found to be more genomically unstable than tumors without such genetic alterations.

Genomic instability caused by *BRCA1/2* deficiency leads to a higher neoantigen load and tumor mutational burden (TMB), which constitute an immune activation signature with a higher level of tumor infiltrating lymphocytes (TILs). This tumor inflammation signature is often counterbalanced by a higher expression of counter-regulatory checkpoint proteins, such as programmed death ligand-1 (PD-L1), to evade immune attack [38].

BC associated with *BRCA1* and *BRCA2* germline mutations is mostly TNBC [39,40], with high mutational loads acquired through HRD and with high PD-L1 expression [41]. Therefore, those tumors have been found to be more immunogenic than HRP cancers [42,43]. Moreover, these mutational signatures in cancer cells were identified as a predictor of therapy responsiveness [44,45]. 

The recent approval of both pembrolizumab and atezolizumab in combination with standard chemotherapy for PD-L1 positive, metastatic TNBC represents an important step forward for the use of immune checkpoint inhibitors (ICB) in BC [46,47].

The first line standard of care for metastatic TNBC is the combination of chemotherapy and immunotherapy with nab-paclitaxel and atezolizumab, according to the results of the IMpassion130 trial, with superior results in terms of the PFS (7.5 vs. 5.0 months, HR = 0.62, 95% CI: 0.49 to 0.78; *p* < 0.001) and OS (25.0 vs. 15.5 months, HR = 0.62, 95% CI: 0.45 to 0.86) in comparison to nab-paclitaxel plus a placebo [47]. These results refer to patients whose tumor expressed the PD-L1 at a level higher than 1%, which represents approximately 40% of all metastatic TNBC.

As revealed by subsequent biomarker analysis, the number of patients with g*BRCA1/2* PV/LPVs enrolled in the IMpassion130 trial was 89 out of 612. In this substudy, immune biomarkers (PD-L1 expression on immune cells and tumor cells, intratumoral CD8, stromal TILs) and germinal *BRCA1/2* alterations were evaluated for association with clinical benefit with atezolizumab and nab-paclitaxel. This analysis revealed that a clinical advantage was only observed in patients whose tumors express PD-L1 on immune cells, regardless of the germinal *BRCA1/2* status [48].

Considering that tumors with deleterious *BRCA1/2* mutations are expected to be genomically unstable, with elevated TMB and a high inflammatory microenvironment, a better response to immunotherapy would have been expected, contrary to what the IMpassion130 analysis showed. Other previous studies failed to demonstrate a relevant clinical benefit in patients with metastatic TNBC treated with ICB, as the KEYNOTE-012 [49] and the KEYNOTE-119 [50], which assessed the role of pembrolizumab in metastatic TNBC, underscoring the heterogeneity of *BRCA1/2*-deficient breast cancers with respect to immunogenicity.

Genomic signatures that might predict immunogenicity in g*BRCA1/2*m BC have been extensively studied, leading to the demonstration that immune gene expression ranges widely among these tumors [43,51].

First of all, BC carries an intermediate TMB compared to cancers in which immunotherapy is widely used, with a median of 2.63 mutations per megabase (mut/MB) among all BCs, compared to 7.2 mut/MB in lung cancer and 13.5 mut/MB in melanoma, as previously reported [52,53]. In general, cancers with high TMB also carry higher TIL counts, higher expression of immune gene signatures and substantial survival benefits from anti-PD1 therapies. 

Moreover, it has been demonstrated that highly immunogenic features (strongest evidence of CD8-driven T cell responses, higher TGF-beta signaling, type I Interferon signaling, NFκB activation) were elevated across HRD-low relative to HRD-high breast tumors. Thus, *BRCA1/2*-related breast cancers represent another example in which the mutational burden and T cell responses are not linked [54]. In contrast, tumor intrinsic features that regulate immune response and suppression are increasingly appreciated as driving forces [55].

HRD-low breast tumors appear to be associated with a pro-inflammatory signature, a predictor of good response to immunotherapy, as opposed to HRD-high disease. This could explain why high-HRD, a frequent feature in *BRCA1/2* mutated tumors, seems to not confer an advantage regarding a response to immunotherapy [56].

### 3.2. Antibody–Drug Conjugates

Another class of drugs that have demonstrated meaningful clinical activity in metastatic TNBC is antibody–drug conjugates (ADCs). Considering the outstanding results that this class of drugs has obtained in HER2+ BC, mainly with the use of trastuzumab emtansine (T-DM1) [57] and trastuzumab deruxtecan (T-Dxd) [58], the search for targetable molecules in TNBC has led to the development of sacituzumab govitecan (SG), an ADC targeting trophoblast cell-surface antigen-2 (Trop-2) conjugated via a cleavable linker to the active metabolite of irinotecan (SN-38). Trop-2 is highly expressed in BC, as well as in most epithelial carcinomas [59,60,61]. Preclinical studies have shown that Trop-2 promotes cell proliferation, inhibits apoptosis, accelerates cell cycle progression and favors tissue invasion and metastasis [62]. The low Trop2 expression in healthy tissues makes it a suitable target for the development of ADCs. The cleavable linker ensures that the cytotoxic molecule is also effective on the neighboring Trop2 negative cells through the “bystander effect”. SG is FDA and EMA-approved for pretreated metastatic TNBC, based on the results from the phase III ASCENT study, where a benefit in terms of the PFS (5.6 vs. 1.7 months, HR = 0.41, 95% CI: 0.32 to 0.52; *p* < 0.001), OS (12.1 vs. 6.7 months, HR = 0.48, 95% CI, 0.38 to 0.59; *p* < 0.001) and ORR (35% vs. 5%) compared to single-agent chemotherapy of the investigator’s choice was demonstrated [63]. 

Exploratory biomarker analysis of this study was conducted in order to evaluate the association of Trop-2 expression and germline *BRCA1/2* status with clinical outcomes. In this analysis, no difference in SG efficacy was seen between g*BRCA1/2*m and g*BRCA1/2*wt patients, and only Trop-2 expression correlated with improved clinical outcomes, with numerically higher efficacy outcomes in the high and medium Trop-2 expression subgroups. However, a trend in favor of improved OS with SG in patients with g*BRCA1/2* P/LP variants compared to g*BRCA1/2*wt patients was observed (15.6 vs. 10.9 months) [64]. Considering the small number of g*BRCA1/2*m patients and the advanced and highly pretreated setting of patients in this study, it is not possible to draw firm conclusions on the predictive value of germline *BRCA1/2* PVs in metastatic TNBC treated with SG. Moreover, all the patients with g*BRCA1/2* P/LP variants enrolled in this trial had previously been treated with a PARPi, which might have had a positive impact on the subsequent outcome of these patients.

It has been demonstrated that PARPi can enhance the activity of topoisomerase-I inhibitors, and in preclinical studies, the combination of SG and PARPi resulted in synergistic growth inhibition compared to SG monotherapy, both in the g*BRCA1/2*m and in the *BRCA1/2* wild-type TNBC. The combination was also well tolerated [65]. Based on these results, a phase I/II study of SG plus talazoparib in metastatic TNBC is currently ongoing (NCT04039230), and subsequent stratification according to the germline *BRCA1/2* status would be needed in order to understand whether germline *BRCA1/2* PVs are predictive of therapy response.

## 4. HER2-Positive Breast Cancer

HER2 protein overexpression/gene amplification, meaning a staining of 3+ in immunohistochemistry (IHC) or gene amplification by fluorescence in situ hybridization (FISH), occurs in 15–20% of primary breast tumors and is associated with decreased disease-free and overall survival [66]. HER2+ BC is rare among *BRCA1/2* mutation carriers; a low frequency (2.1% to 10%) of HER2+ status and *BRCA1* PVs/LPVs carriers and a slightly higher rate (6.8% to 13%) in those with germinal mutations in *BRCA2* have been found [67]. Recently, an observational study was conducted in order to evaluate the prognostic significance of germline *BRCA1/2* PVs in patients with HER2-positive BC [68]. However, this study assessed patients with early or locally advanced BC (stage I to IIIA), and data on metastatic disease are missing. In this study, an interaction between *BRCA1/2* PVs and HER2-positive status was found to correlate with worse survival after adjusting for prognostic variables. This study provides evidence for the first time that the co-occurrence of *BRCA*1/2 mutations and HER2-positive status is a poor prognostic factor in patients with early or locally advanced breast cancer. Notably, we observed that HER2+/g*BRCA1/2*m cases had a poorer 5-year OS rate than the controls (HER2+/g*BRCA1/2*wt and HER2−/g*BRCA1/2*m and HER2−/g*BRCA1/2*wt). In preclinical models, inactivating *BRCA2* mutations correlated with a response to the HER2 tyrosine kinase inhibitors tucatinib and neratinib. Furthermore, the addition of olaparib enhanced the effect of neratinib in breast cancer cell lines, and niraparib enhanced the neratinib effectiveness in ovarian cancer [69]. The finding that the co-occurrence of *BRCA1/2* mutations and a HER2-positive status is associated with worse OS in patients with early or locally advanced breast cancer may be proof of concept that a combined pharmacological intervention directed to these targets could be synergistic. Clinical trials evaluating novel combinations of PARPi plus anti-HER2 therapies are warranted in this setting. In particular, a phase II trial (NCT03931551) aimed to recruit HER2+ BC patients with gene alterations in the HRR DNA pathway (including germline deleterious mutations in *BRCA1* or *BRCA2* genes) in order to evaluate the efficacy of the association of olaparib and trastuzumab. However, this study was recently terminated because of very slow recruitment.

Finally, although not selectively designed for the cohort of *BRCA1/2* mutated BCs, an interesting phase I trial (NCT04585958) is enrolling patients with HER2+ solid tumors in order to evaluate the safety and tolerability of the combination of trastuzumab deruxtecan and olaparib. 

## 5. HR-Negative, HER-2 Low Breast Cancer

Nowadays, a new subtype of BC deserves special mention: the HR-negative HER2-low (HR− HER2-low). HER2-low BC (irrespective of HR status) has gained more and more interest with the results described with T-DXd, a novel ADC targeting HER2. Of note, this compound showed relevant activity also in HER2-low patients (defined as FISH negative, but IHC 1+ or 2+) [70]. According to this definition, a considerable percentage of BCs that so far have been classified as triple-negative can now be included in the HR-negative HER2-low subgroup. This means that almost all TNBC (with the exception of the IHC 0 tumors) could now benefit from treatment with a drug that is active against the HER2-low disease. Moreover, this patient population of HR- HER2-low, hitherto classified as TNBC, is the one with a higher probability of harboring *BRCA1/2* germline mutations. 

The DESTINY-Breast01 and DESTINY-Breast04 subgroup analysis and the biomarker evaluation regarding g*BRCA1/2* status are missing; therefore, there is no data about the efficacy of T-DXd in patients with HER2-low ABC and g*BRCA1/2* P/LP variants. 

## 6. Conclusions

Information on the outcomes in patients with ABC and germline *BRCA1/2* mutations are very limited, and only a few trials included these patient populations in their subanalyses [47,63] or performed retrospective analyses [22,23,24] (Figure 1 and Table 1).

Efficacy results stratified by *BRCA1/2* status were reported in metastatic TNBC treated with a combination of atezolizumab and nab-paclitaxel or with sacituzumab govitecan, and no significant differences were observed. The evaluation of the response to T-DXd in HER2-low ABC BRCA1/2 carriers is missing but is strongly encouraged given the proportion of tumors nowadays classified as HER2-low.

An important area of research is represented by combination trials of PARPi with other biological agents (CDK4/6 inhibitors, PI3K inhibitors, immune checkpoint blockers, antibody–drug conjugates), either in patients with germline and/or somatic *BRCA1/2* PVs/LPVs and in *BRCA1/2* wild-type patients. In particular, results are needed for those drug combinations for which the preliminary data demonstrated synergistic antitumor activity, such as for olaparib plus alpelisib in breast and ovarian cancer or sacituzumab govitecan plus talazoparib in mouse models. 

The combination of PARPi, the milestone of targeted therapy in g*BRCA1/2*m ABC, with other targeted agents represents a frontier for the future, also taking into account a biological rationale for trying to overcome the resistance to PARPi. 

With increasing access to germline genetic testing and as indications for patients expand, particularly for therapeutic purposes in both early and advanced breast cancer, the *BRCA1/2* status will be available to more and more patients, and the number of patients with germline pathogenic mutations will be higher than in the past. Therefore, the integration of germline genetic test information into new trial designs and subgroup analyses is required. Thereafter, increasing knowledge about the efficacy of new biological agents in *BRCA1/2* mutated disease will help to optimize both the patient and treatment selection.

## Figures and Tables

**Figure 1 cancers-15-03305-f001:**
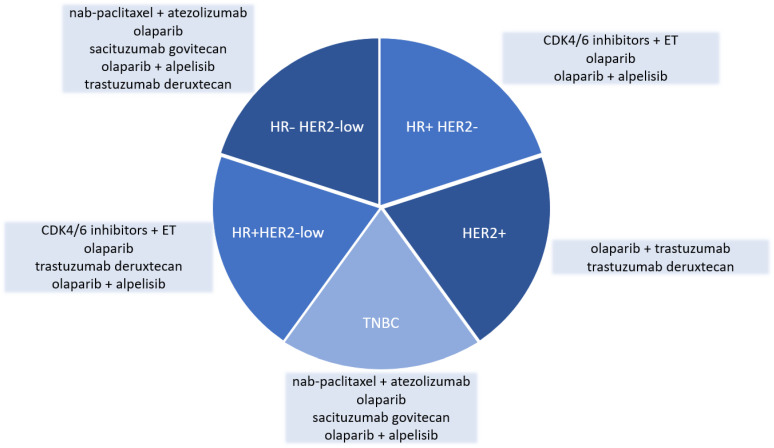
Summary of agents discussed across different tumor subtypes. HR: hormone receptor, HER2: human epidermal growth factor receptor 2, TNBC: triple-negative breast cancer, CDK4/6: cyclin-depended kinase 4/6, ET: endocrine therapy.

**Table 1 cancers-15-03305-t001:** Evidence presented, current clinical practice and ongoing clinical trials discussed.

		Evidence in g*BRCA1/2*m	Practice	Clinical Trial
HR+ HER2−	Olaparib	**✓**	**✓**	NCT02000622
	Talazoparib	**✓**	**✓**	NCT01945775
	CDK4/6i + ET	**?**	**✓**	NCT01740427 NCT01958021 NCT02278120 NCT02246621
	CDK4/6i + olaparib + ET	** X **	** X **	NCT03685331 ongoing
	PI3Kαi + ET	** X **	** X **	NCT02437318
	PI3Kαi + olaparib	**?**	** X **	NCT01623349
TNBC	Nab-paclitaxel + atezolizumab	**?**	**✓**	NCT02425891
	Sacituzumab govitecan	**?**	**✓**	NCT02574455
	Sacituzumab govitecan + talazoparib	** X **	** X **	NCT04039230 ongoing
HER2+	Olaparib + trastuzumab	** X **	** X **	NCT03931551
	Trastuzumab deruxtecan	**?**	**✓**	NCT03248492 NCT03734029

HR: hormone receptor, HER2: human epidermal growth factor receptor 2, TNBC: triple-negative breast cancer, CDK4/6i: cyclin-depended kinase 4/6 inhibitors, ET: endocrine therapy, PI3Kαi: phosphoinositide 3-kinase alpha inhibitors.

## Data Availability

The data presented in this study are openly available at PubMed Central (PMC).

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
