# Peer review of "Beyond PARP Inhibitors in Advanced Breast Cancer Patients with Germline BRCA1/2 Mutations: Focus on CDK4/6-Inhibitors and Data Review on Other Biological Therapies"

_cancers, 2023, doi:10.3390/cancers15133305_

Round 1

Reviewer 1 Report

Dear Authors, 

I reviewed with interest the paper entitled “Biological Therapies Beyond PARP-Inhibitors in Advanced 2 Breast Cancer Patients with Germline BRCA1/2 Mutations”.

First, I would strongly congratulate with the authors for their hard work for the present review, which covers an actual and very interesting topic such as summarizing the available knowledge on new generation drugs across different breast cancer isubtypes caused by germline pathogenic variants in BRCA1/2 genes. 

I found the present study interesting and well written - no major concerns with language editing or general fluency.

I found the present study interesting, well written and fluent to read. It concerns with an actual topic, which is of major importance.

The title is descriptive of what authors have explored in their work. The background and scientific rationale for carrying out the study are well presented. Tables and Figures are clear and not repetitive, as well as Results section. The paper results methodologically correct. Discussion is adequately implemented with the relevant literature, and conclusions are well stated and justified by results. I have no concerns or suggestions.

Author Response

Thank you for you kind comments.

Reviewer 2 Report

1.The name is generalized, and data is only presented for CDK4/6 inhibitors-consider editing.

2.The data for CDK4/6 inh in BRCA PV carriers is only based on 85 patients in ref. 22 "Of 2968 patients with HR+/HER2- mBC receiving a CDK4/6 inhibitor, 859 (28.9%) had known gBRCA status, of whom 9.9% were gBRCAm"- Cited wrongly in lines 107-8., 10 pt. in ref.23 and 6 pt. in ref. 24. All are retrospective analysis.

3.For all other drugs the discussion may be interesting but actual data regarding activity in BRCA PV carriers is not yet available....

Author Response

1.The name is generalized, and data is only presented for CDK4/6 inhibitors-consider editing.

Response 1: Data on sacituzumab govitecan and atezolizumab are also presented, although not significant in terms of differences in outcome for patients with germline mutations in BRCA1/2.

2.The data for CDK4/6 inh in BRCA PV carriers is only based on 85 patients in ref. 22 "Of 2968 patients with HR+/HER2- mBC receiving a CDK4/6 inhibitor, 859 (28.9%) had known gBRCA status, of whom 9.9% were gBRCAm"- Cited wrongly in lines 107-8., 10 pt. in ref.23 and 6 pt. in ref. 24. All are retrospective analysis.

Response 2: Thank you, we will provide appropriate corrections

3.For all other drugs the discussion may be interesting but actual data regarding activity in BRCA PV carriers is not yet available...

Response 3: Thank you, we wanted to emphasise the increasing importance of including in the near future patients' genetic data into clinical trials.

Round 2

Reviewer 2 Report

ok

Author Response

Thank you